# Appraisal of Experimental Methods to Manage Menopause and Infertility: Intraovarian Platelet-Rich Plasma vs. Condensed Platelet-Derived Cytokines

**DOI:** 10.3390/medicina58010003

**Published:** 2021-12-21

**Authors:** E. Scott Sills, Samuel H. Wood

**Affiliations:** 1Regenerative Biology Group, FertiGen/CAG, San Clemente, CA 92673, USA; 2Palomar Medical Center, Department of Obstetrics & Gynecology, Escondido, CA 92029, USA; 3Gen 5 Fertility Center, San Diego, CA 92121, USA

**Keywords:** platelets, cytokines, angiogenesis, embryo, menopause, fertility

## Abstract

The first published description of intraovarian platelet-rich plasma (PRP) appeared in mid-2016, when a new experimental technique was successfully used in adult human ovaries to correct the reduced fertility potential accompanying advanced maternal age. Considering the potential therapeutic scope of intraovarian PRP would likely cover both menopause and infertility, the mainstream response has ranged from skeptical disbelief to welcome astonishment. Indeed, reports of intraovarian PRP leading to restored menses in menopause (as an alternative to conventional hormone replacement therapy) and healthy term livebirths for infertility patients (from IVF or as unassisted conceptions) continue to draw notice. Yet, any proper criticism of ovarian PRP applications will be difficult to rebut given the heterogenous patient screening, varied sample preparations, wide differences in platelet incubation and activation protocols, surgical/anesthesia techniques, and delivery methods. Notwithstanding these aspects, no adverse events have thus far been reported and ovarian PRP appears well tolerated by patients. Here, early studies guiding the transition of ‘ovarian rejuvenation’ from experimental to clinical are outlined, with mechanisms to explain results observed in both veterinary and human ovarian PRP research. Current and future challenges for intraovarian cytokine treatment are also discussed.

## 1. Introduction

Platelet-rich plasma (PRP) represents a physiologic signaling aggregate comprising hundreds of platelet derived cytokines obtained from blood samples, collected by standard venipuncture [1]. As a refinement of conventional PRP, growth factors of platelet origin may be further processed to enrich this releasate after activation [2]. Interest in PRP applications has grown over the past 15 years, and since 2016 it has attracted particular attention in experimental reproductive biology. Understandably, the claim to ‘rewind the biological clock’ [3] has been cautiously received. It is not known how many IVF clinics now offer ‘ovarian rejuvenation’ although it is a safe assumption the number was zero prior to 2016. In contrast, considerable experience with autologous PRP (fresh and frozen) has already been reported in cardiothoracic surgery [4], scalp hair regrowth [5], dermatology [6], oral surgery [7], sports medicine [8] and other clinical fields [9]. While the absence of randomized placebo-controlled clinical trials regarding intraovarian PRP must be acknowledged, this deficiency did not block IVF from entering mainstream fertility practice without RCT support [10].

Rejuvenation arrives on the gynecology stage with its (non-pharmacologic) promise to improve ovarian function [11]. However, much remains to be discovered regarding exactly what intraovarian PRP and/or condensed plasma cytokines does within reproductive tissue to achieve this. It is known that ovarian perfusion corresponds closely with intrafollicular oxygenation, and those follicles with the lowest dissolved oxygen levels most often provide oocytes with cytoplasmic and chromosomal error [11]. Unfortunately, when this imbalance remains uncorrected, embryos least likely to implant originate from follicles which showed impaired vascularity [11]. ‘Ovarian rejuvenation’ could thus ameliorate a locally hypoxic microenvironment, or perhaps supply platelet-derived signaling necessary to induce de novo oocyte differentiation.

Autologous PRP has been used occasionally as an intrauterine lavage, aiming to boost endometrial receptivity and enhance embryo implantation [12]. In the follicular recruitment IVF space, intraovarian PRP joins a crowded cast of newer interventions such as human growth hormone, aspirin, heparin, DHEA, antioxidants, and screening hysteroscopy [13]. However, what research exists to support proposed PRP pathways leading to ovarian rejuvenation?

## 2. Therapeutic Rationale

Platelets contain multiple granules which, upon activation, deliver numerous cargo proteins including platelet-derived growth factor (PDGF), fibroblast growth factor (FGF), vascular endothelial growth factor (VEGF), epidermal growth factor (EGF), transforming growth factor-beta 1 (TGF-β1), insulin-like growth factor (IGF), connecting tissue growth factor (CTGF), hepatocyte growth factor (HGF), and others [9,11]. The roster of releasate contents seems ever growing; these moieties orchestrate cellular growth and direct repair following tissue injury. For older or impaired ovarian tissue, small series and case report data suggest this signaling milieu can contribute to improved stromal perfusion, enable an enlarged follicle pool, recruit latent oocytes, and produce healthy term livebirths [14,15].

While intraovarian PRP is usually regarded as a precursor to IVF, the ‘reset’ established after platelet cytokine treatment can confer benefits even if not followed by such complex treatments [15,16,17]. This could occur due to temporary resolution of tissue dysfunction associated with hypoperfusion characteristic of the senescent ovary [11]. 

## 3. Ovarian PRP: Veterinary and Human Research

What might PRP accomplish in the setting of impaired or even obliterated reserve as with menopause? This question was explored in an animal model where intraperitoneal 4-vinylcyclohexene dioxide administration was used as a gonadotoxin for total ovarian collapse. Next, rat intraovarian PRP injections were followed by documentation of cellular changes as well as expression of angiogenic-related transcripts *ANGPT2* and *KDR* via real-time qPCR. While histopathological review confirmed an ovarian insufficiency (POI) state after initial conditions, injection of PRP substantially reduced the extent of follicular atresia and inflammatory response [18]. An uptick was also measured in *ANGPT2* and *KDR* transcript expression in POI rats secondary to enhanced inflammation, but reduced after PRP administration vs. controls. Perhaps most crucially, improvement in litter counts was documented among animals receiving PRP compared to the non-treated POI group [18]. This study also found that intraovarian PRP also protected morphologically normal follicles from degeneration. This parallels earlier observations among rats with experimentally induced PCOS, where improved ovarian antioxidant potential and enhanced follicular development after PRP mitigated deleterious oocyte effects in PCOS [19].

Research from Milan [20] described bovine response to administration of autologous intraovarian PRP before programmed superovulation. A significant improvement was noted in mean follicle number between control vs. PRP injected ovaries, and significantly more high-grade blastocysts were generated following PRP use [20]. Additional animal experiments have found a beneficial (rescue) effect of PRP on ovarian function in female rats with ovarian damage induced by cyclophosphamide, concluding this approach could lead to improved primordial, primary, secondary, and antral follicle counts [21].

## 4. Patient and Protocol Differences

Both platelet concentration and derivative cytokine releasate show considerable individual variation [22], and this aspect of PRP treatment is important to review with potential patients before ovarian rejuvenation is attempted. At least two techniques exist to process PRP (see Figure 1), although ‘best practice guidelines’ are not yet available to help screen patients or suggest a specific therapy. While a minimum platelet level necessary to elicit a regenerative response probably depends on the intended target tissue, animal research [23] found a threshold approaching 1M platelets/mL (i.e., a 3–8 fold enrichment over baseline) as sufficient for bone healing. Of note, these experts did acknowledge a ‘more is not always better’ paradigm [23] to highlight the need for additional research.

Prospective intraovarian PRP human data (*n* = 182) identified significant differences in baseline platelet count among responders vs. non-responders [24], validating a relation between platelet count and subsequent ovarian reserve post-treatment. Even for women with platelet levels on the lower margin of normal, it might be possible to collect two samples within a few hours to pool autologous blood for aggregate processing and same day (fresh) injection. By facilitating ovarian PRP treatment with an augmented platelet concentration, this may overcome marginally low platelet counts which would otherwise be disqualifying. For cases with absolute thrombocytopenia (PLT < 100 K), formal hematology consult is appropriate as these patients would be high risk for IVF and pregnancy.

The impact of handling technique on platelet lysate has also been studied [25], finding that the method of sample preparation can significantly influence the growth factor profile. Specifically, platelet-derived cytokine levels were markedly increased in non-calcium activated PRP with a freeze–thaw–freeze incubation; a major disadvantage was also reported if room temperature incubation was used [25]. 

Lower PRP concentrations might still block nasoseptal cells from losing their chondrogenic potential due to in vitro expansion, thereby promoting recommitment [26]. Reduced concentration of platelet releasate was able to augment mesenchymal stromal cells as noted by upregulated gene expression, sulfated glycosaminoglycan production, and compressive modulus after in vitro culture. Some markers of regenerative action were again impaired at higher concentration of platelet releasate (10%), emphasizing the need to define an optimal sample preparation method [26].

Platelet-derived cytokines have been quantified following activation either with calcium alone or calcium/thrombin [27]. High concentrations of platelet-derived growth factor, endothelial growth factor, and transforming growth factor (TGF) were secreted with interleukin (IL)-4, IL-8, IL-13, IL-17, tumor necrosis factor (TNF)-α and interferon (IFN)-α. Unsurprisingly, no cytokines were secreted without platelet activation. TGF-β3 and IFNγ were absent in all studied fractions. Clots obtained after platelet coagulation retained a high cytokine concentration, including platelet-derived growth factor and TGF [27].

Research from colleagues in Ukraine reported on 38 women with low ovarian reserve and at least two failed IVF cycles (age 31–45 yrs) who underwent ovarian PRP [28]. In their experience, route of PRP administration was either via laparoscopy or transvaginal ultrasound, and patients were monitored over one year. Significant improvement in ovarian function was noted after treatment, including 10 pregnancies and delivery of six healthy babies [28]. The largest single-center experience with intraovarian PRP probably remains at Genesis Athens (Greece) [1], where this team actively publishes results on specific patient groups. For example, among menopausal women receiving ovarian PRP, 24 of 30 attained restored menstruation and improved hormonal levels/ovarian antral follicle count—sometimes as soon as one month after injection [1]. This finding extends results from an earlier questionnaire study (*n* = 80) on quality of life/non-reproductive outcomes where ovarian PRP was administered as an alternative to standard HRT [29]. Due to loss of follow-up after ‘menopause reversal’, no longitudinal data were available on this group to determine duration of treatment effects. Additionally, in 2019, ovarian PRP data were published from a prospective matched cohort study where selected reproductive outcomes were tracked in 20 IVF patients receiving this treatment vs. 20 control IVF patients without ovarian PRP. In this pilot trial, a trend towards improved embryo implantation and livebirth rate was measured among IVF patients who received ovarian PRP [30]. One technique (Segova) mentioned by experts in Serbia reports on a PRP processing method using ‘special systems and machines’ to increase growth factor levels up to 18 times the initial concentration [31]. 

## 5. Considerations and Contraindications

Before enrolling patients for ovarian PRP or intraovarian injection of condensed plasma cytokines, the same baseline considerations for IVF or HRT should apply. Ovarian PRP patient entry criteria followed during the NIH Clinical Trial [NCT03178695] included patients with at least one ovary, infertility of >1yr duration, at least one prior failed (or canceled) IVF cycle, or amenorrhea for at least three months. However, patients who are otherwise healthy, but have undetectable reserve (serum AMH < 0.03 ng/mL) and considered so unsuited for fertility treatment that they were never allowed to try IVF with native oocytes elsewhere, should not be excluded [32]. At exam, it is important to confirm safe ovarian access via transvaginal ultrasound prior to ovarian PRP. For those where background medical conditions are uncertain, medical clearance is required. Patients with ongoing pregnancy, current or previous IgA deficiency, ovarian insufficiency secondary to sex chromosome etiology, prior major lower abdominal surgery resulting in pelvic adhesions, anticoagulant use for which plasma infusion is contraindicated, psychiatric disorder precluding study participation (including active substance abuse or dependence), obesity, current smokers, ongoing malignancy, or chronic pelvic pain should be excluded [24].

During pre-treatment screening, it is important to query aspirin/NSAID use, as agents in this class will attenuate platelet function. Specifically, irreversible inhibition of platelet COX-1 by aspirin suppresses precursors required for downstream cytokine signaling [33]. Recent research has clarified the mechanisms involved in aspirin’s brake effect on platelet activation [34] and since platelet activation is essential for ovarian PRP to achieve any therapeutic gain, requiring a NSAID-free window of at least 10d before planned intraovarian injection seems reasonable. Likewise, for patients taking pentoxifylline, this medication also merits caution in advance of ovarian PRP, as it impairs transforming growth factor-beta and platelet-derived growth factor production [35]. Pentoxifylline can also block platelet-associated cytokine release in some settings [36] and should be avoided to optimize overall platelet functional potential.

Pre-intake considerations notwithstanding, a published opinion accurately identified weaknesses in ovarian PRP reports currently available [37]. Particularly noted were the paucity of delivery data after ovarian PRP, the heterogeneity of commercial systems used in plasma processing, and absence of pre- vs. post-PRP antral follicle count [37]. Regarding the first critique, the nascent phase of autologous plasma factor research explains why delivery data remain confined to small series and case reports. Highlighting the underdeveloped state of ovarian PRP is proper and the call for delivery rate information is a necessary message. This is a familiar deficiency, and warrants acceptance against a larger unresolved debate about how best to report ‘success’ for IVF in general [38,39]. The second point on differences in PRP sample handling, processing, and administration is also compelling, and represents a serious hurdle to be cleared if usable comparisons are to be delivered. Few published PRP protocols use the exact same kit for specimen collection and processing, centrifugation ‘spin’ parameters, or which reagent is used for platelet activation. Such differences can impact substrate platelet concentration, its cytokine profile, and efficiency of growth factor release. Normal temporal and biological factors can influence platelet availability and make assessments across multiple PRP platforms difficult to compare [9,40]. This variation with ovarian PRP methods presents problems for meaningful cross-center comparisons, yet with description of sample preparation, surgical approach, and full reporting of findings, it could actually accelerate better understanding of which PRP technique works best for patients. Concerning antral follicle count data to score response to intraovarian PRP, collecting this information would probably add little to ovarian rejuvenation given its limited reproducibility and low measurement consistency secondary to operator and/or ultrasound equipment variation. Serum AMH, in contrast, is much easier to standardize, and thus represents a preferred marker for follicular potential/ovarian function [41].

Reliable measurement of selected constituent molecular signals derived from platelets as a function of activation reagent and in vitro incubation method can offer descriptive information for reproductive biology. As multiple ways exist to perform ‘ovarian rejuvenation’, it will be useful to document differences in cytokine concentrations by technique. Case report and small series reports, while interesting, are most beneficial for the possible rather than the probable. It has been acknowledged that many IVF techniques now accepted as routine clinical practice once were experimental with humble or obscure origins. As noted here, the safety and efficacy of such treatments should be supported by data ideally from multiple randomized clinical trials [42]. We agree with Kamath et al. [13], as caution is appropriate where use of early, investigational techniques is proposed.

## 6. Conclusions

The central question of whether or not the adult human ovary retains the capacity to produce *de novo* oocytes remains open. In postnatal CNS tissues also once thought irreplaceable, a similar issue is found regarding functional recovery and cellular regeneration [43,44]. Therefore, under certain conditions, the key objective of cellular recovery here likewise may need reconsideration. Working with a murine eye model, upregulation of specific genes has restored a ‘youthful’ DNA methylation pattern as well as axonal regrowth following tissue damage—an approach enabling vision to return after blindness injury in mice [45]. Relevant genes implicated in this transcriptome expression include Sox2, Oct4, and Klf4 [45]. Of note, human platelet lysate has been shown to promote mRNA expression of such ‘mitotic bookmarking factors’ [46]. Building on such studies, reproductive biology can gain much to determine if these (or other) signals are operant in postnatal ovarian function. Small series and case data now exist to describe reproductive outcomes after intraovarian PRP or platelet-derived cytokines. While experience with serum AMH (as an estimate of ovarian reserve) following autologous ovarian PRP requires multicenter validation, additional research to characterize specific PRP cytokine components will be even more useful. 

Observations discussed here from clinical work in ovarian rejuvenation favor a hypothesis for derivative neovascularity to modulate oocyte competence, by augmenting cellular oxygenation and/or reducing levels of intraovarian reactive oxygen species [11]. Related experimental animal research found ovarian function and follicular development were indeed promoted after VEGF-mediated vascular remodeling [47].

The need for rigorous RCT data on intraovarian insertion of platelet-derived cytokines before this innovation enters accepted IVF practice should be viewed as familiar terrain for reproductive medicine [10,48]. Declining ovarian reserve and ineffective fertility responses have become more formidable with advanced maternal age and cannot be defeated by gonadotropins alone. Similarly, perimenopause marks a symptomatic decay in female sex hormone output which at present is usually managed by exogenous hormone replacement therapy [49,50]. For both patient populations, the prospect of effective ‘ovarian rejuvenation’ would hold considerable appeal. Could autologous platelet cytokines help meet this need? While a postnatal folliculogenesis model might explain improved ovarian function among older women, important challenges remain [51,52]. A population of adult ovarian germline stem cells might be latent—but available—for differentiation, the exact process through which cytokines induce such development requires additional study. What has been suggested from early reports on platelet cytokines suggests these moieties can initiate morphogenetic processes normally seen during evolutionary development [53]. Indeed, recent observations in two species of sacoglossan snail (gastropoda) have demonstrated extreme regeneration, wherein a severed head was able to regrow an entire new body within approximately 20d [54]. Unless a parallel process can be discovered to achieve limited postnatal replenishment of the human oocyte pool, reliance on IVF with donor oocytes will continue. Meanwhile, characterization of the activated PRP substrate, its derivative growth factor profile, receptor targets, optimal sample delivery, and ideally RCT data to support this treatment are still needed.

## Figures and Tables

**Figure 1 medicina-58-00003-f001:**
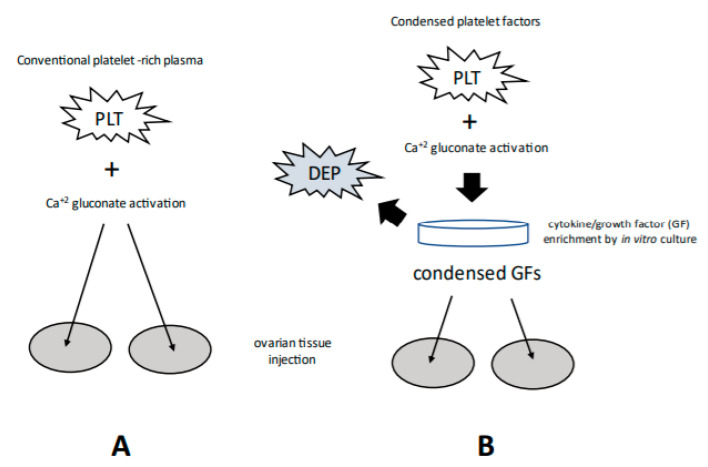
At least two methods of PRP sample preparation are currently in use, including conventional activation (**A**) and condensed cytokines isolated after in vitro platelet (PLT) incubation/processing (**B**). Note that depleted platelets (DEP) are removed (in B) following concentration of platelet releasate. Although bioactivity for both is a function of multiple signaling moieties, the concentration of such growth factors should be markedly increased along path B. Relevant platelet-derived cytokines include Vascular endothelial growth factor (VEGF), a signal protein promoting angiogenesis; Ligand of CD40 (CD-40L), an inflammatory signal for endothelium, platelets, and leukocytes; Interleukin-1β (IL-1β), an inflammatory marker involved in cell growth, differentiation; Interleukin-8 (IL-8), which initiates angiogenesis, perfusion, and movement to injury/infection sites; PLT derived growth factor (PDGF), essential for vascular development, proliferation of fibroblasts, osteoblasts, tenocytes, vascular SMCs and mesenchymal stem cells; and PLT factor 4 (PF4), central in organizing platelet aggregation.

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
