# Peer review of "Appraisal of Experimental Methods to Manage Menopause and Infertility: Intraovarian Platelet-Rich Plasma vs. Condensed Platelet-Derived Cytokines"

_medicina, 2021, doi:10.3390/medicina58010003_

Round 1

Reviewer 1 Report

General comments:

The authors have reported a very comprehensive review article about a hot topic that has been discussed in the literature.

Title

The title represents the manuscript’s contents.

Abstract:

Abstract section has been designed according to the journal style.

Introduction:

The topic has been introduced properly.

Therapeutic rationale, Ovarian PRP: Veterinary and human research, Patient and protocol differences, Considerations and contraindications:

The authors have reviewed almost all the published literature and have described the data in detail in all these sections.

Conclusion:

The authors have finalized the literature review in this section.

References:

Most recent and actual references have been cited.

Author Response

Thank you for this review - no changes required.

Reviewer 2 Report

In this manuscript authors have summarized the methods to manage menopause and infertility which specific focus on intaovarian injection of PRP vs. condensed platelet-derived cytokines.

In my opinion the manuscript is well structured and very well written

Author Response

(The authors gave the same response as above.)

Reviewer 3 Report

In this manuscript, Sills and Wood evaluated comprehensively a new experimental technique (intraovarian platelet-rich plasma, PRP) in improving menopause and infertility. The topic is very interesting, far reaching, and significantly advance the field.  Therapeutic rationale, researches from both animal and human, protocol differences, and potential contraindication are all covered and explained well in text. The work will be helpful for clinical assessment and therapy of certain infertile/subfertile women. Altogether, the manuscript is well written and acceptable to be published from this reviewer, with only one point that is recommended to be addressed as below.

Authors might consider expanding the introduction section, with more background knowledge, given that this is a new experimental technique to certain field. That would also be more friendly and readable for broad readers.

Author Response

Thank you for this guidance - additional background on PRP has now been added at lines 44-52, as suggested.